# Protein flexibility is required for vesicle tethering at the Golgi

**Pak-yan Patricia Cheung[1], Charles Limouse[2], Hideo Mabuchi[2], Suzanne R Pfeffer[1]\***

[1]Department of Biochemistry, Stanford University School of Medicine, Stanford, United States; [2]Department of Applied Physics, Stanford University, Stanford, United States

**Abstract** The Golgi is decorated with coiled-coil proteins that may extend long distances to help vesicles find their targets. GCC185 is a trans Golgi-associated protein that captures vesicles inbound from late endosomes. Although predicted to be relatively rigid and highly extended, we show that flexibility in a central region is required for GCC185's ability to function in a vesicle tethering cycle. Proximity ligation experiments show that that GCC185's N-and C-termini are within <40 nm of each other on the Golgi. In physiological buffers without fixatives, atomic force microscopy reveals that GCC185 is shorter than predicted, and its flexibility is due to a central bubble that represents local unwinding of specific sequences. Moreover, 85% of the N-termini are splayed, and the splayed N-terminus can capture transport vesicles in vitro. These unexpected features support a model in which GCC185 collapses onto the Golgi surface, perhaps by binding to Rab GTPases, to mediate vesicle tethering.

**\*For correspondence:** pfeffer@ stanford.edu

## Introduction

Membrane trafficking involves the collection of cargo into transport vesicles, movement of vesicles along cytoskeletal tracks, and tethering, docking and fusion of vesicles at their target membranes. Tethering is the process by which partner membranes are brought together into close proximity to permit their subsequent fusion (*Pfeffer, 1999*; *Sztul and Lupashin, 2006*; *Yu and Hughson, 2010*). Two classes of transport vesicle tethers have been described to date: multi-subunit complexes such as the Exocyst, COG, Dsl1 and TRAPP complexes (*Bröcker et al., 2010*), and larger, dimeric coiled-coil containing proteins such as p115 and EEA1 (*Sztul and Lupashin, 2006*; *Munro, 2011*). Both classes share the ability to bind to Rab GTPases, SNARE proteins and vesicle coat complexes, suggesting that they play important roles in coordinating molecular events important for target recognition and membrane fusion (*Short et al., 2005*; *Cai et al., 2007*; *Sinka et al., 2008*; *Hayes et al., 2009*; *Hughson and Reinisch, 2010*).

Golgins are long, Golgi-associated proteins that contain a high proportion of sequences that are predicted to form coiled coils (*Munro, 2011*; *Short et al., 2005*). They are anchored to the Golgi via their C-termini, and models suggest that they protrude long distances, relying on the relative rigidity of the coiled coil structure to provide a meshwork that can capture vesicles in the vicinity of the Golgi (*Yu and Hughson, 2010*; *Munro, 2011*). GCC185 is a Golgin required for the transport of mannose 6-phosphate receptors (MPRs) from late endosomes to the trans Golgi network (*Reddy, 2006*; *Derby et al., 2007*). Cells depleted of GCC185 accumulate MPRs in Rab9 GTPase- and AP-1-decorated transport carriers (*Hayes et al., 2009*; *Reddy, 2006*; *Brown et al., 2011*), providing an assay for GCC185 tethering capacity. GCC185 is likely anchored on the TGN surface by cooperative binding of its C-terminus to both Rab6 and Arl1 GTPases (*Burguete et al., 2008*). We (*Reddy, 2006*; *Burguete et al., 2008*) and others (*Derby, 2004*) have shown that GCC185 binds Arl1 rather weakly, but Rab6A binding greatly enhances Arl1 interaction (*Burguete et al., 2008*).

**eLife digest** Some cells release molecules, such as hormones and neurotransmitters, to signal to other cells and influence how they work. As part of the release process, these molecules are packaged into small, balloon-like structures called vesicles. Such vesicles move around within cells and are able to find the right place to release their contents to the outside.

A cellular compartment called the Golgi complex helps to prepare proteins for release from the cell. Vesicles can bind tethering proteins on the surface of the Golgi, but it was not clear how these proteins are able to capture the correct kind of vesicle. The prediction was that the proteins are rigid, shaped like pipe cleaners that stick out from the Golgi as a meshwork that traps vesicles.

Cheung et al. isolated a specific Golgi tethering protein (called GCC185) from cultured human cells and used a technique called atomic force microscopy to visualize its structure. This revealed that this protein is not rod-like; it is instead rather floppy, and has two arms at one end that may 'hug' the incoming vesicle. Cheung et al. showed that this protein needs its middle portion to be floppy to work correctly. This changes the way we think about how vesicles are able to find their corresponding targets on different compartments inside cells.

Further experiments are now needed to answer a number of questions. What does the tether look like when actually bound to a vesicle? What happens after the vesicle binds – how does the tether let go? What other components are needed for vesicle capture and release?

Arl1's role in anchoring GCC185 was questioned because it seemed not to be required if siRNA-depleted by 80% (*Houghton et al., 2009*); in fact, 90% depletion of Arl1 is required to displace GCC185 from membranes and to detect its role in Golgi localization of the GCC185 C-terminus (*Burguete et al., 2008*).

To date, we have little (if any) information, whether Golgins are actually as long as predicted, whether they are relatively rigid, or even where or how they bind transport vesicles. Moreover, once they bind vesicles, what happens next? Many coiled-coil tethers contain putative hinges, located between predicted rod-like, coiled-coil domains (*Barr and Short, 2003*; *Cai et al., 2007*; *Lupas et al., 1991*; *Munro, 2011*; *Pfeffer, 1999*). Do these regions function as hinges? We have taken a multidisciplinary approach to investigate the mechanism of vesicle tethering by a trans Golgi network (TGN)-associated Golgin, GCC185. Our data provide novel insight into the tethering process and suggest that the TGN Golgin, GCC185, uses an unexpected mechanism to accomplish vesicle tethering.

## Results

Secondary structure prediction algorithms (*Lupas et al., 1991*) suggest that GCC185 may be almost entirely coiled-coil, perhaps >200 nm in length. To determine the protein's actual length, we expressed and purified an N-terminally GFP-FLAG-tagged, GCC185 wild type in HEK293 cells. To test models of vesicle tethering that invoke tether bending, we also generated a '△hinge' construct missing two potential hinge domains near the middle of the protein that were selected based upon predicted breaks in coiled coil sequences (*Figure 1*; △751-805, △890-939). The wild type protein migrated at ~220 kD upon SDS-PAGE; GCC185 △hinge was somewhat smaller (*Figure 2A*). Purified GCC185 was imaged by atomic force microscopy (AFM) after application onto a mica surface. AFM of GCC185 under physiological buffer conditions revealed the molecule as a parallel dimer of ~145 nm in length with splayed ends (*Figure 2B,D*; *Figure 3*, *Figure 3—figure supplement 1–3*). The N-terminal, globular GFP moiety could often be visualized; the C-terminal, smaller 'GRIP domain' could also be detected. GCC185 contains numerous cysteine residues (*Figure 3—figure supplement 1,2*); it was important to include dithiothreitol to mimic the reducing environment of the cytoplasm for all experiments.

Eighty five percent of the molecules showed splaying of at least one end of the dimeric coiled coil, and 38% showed splaying of both ends (*Figure 3*). It is important to note that the N-terminal GFP used in AFM experiments did not impede the ability of the N-terminus to form a coiled coil because many fewer splays and central bubbles were seen in initial experiments when GCC185 was

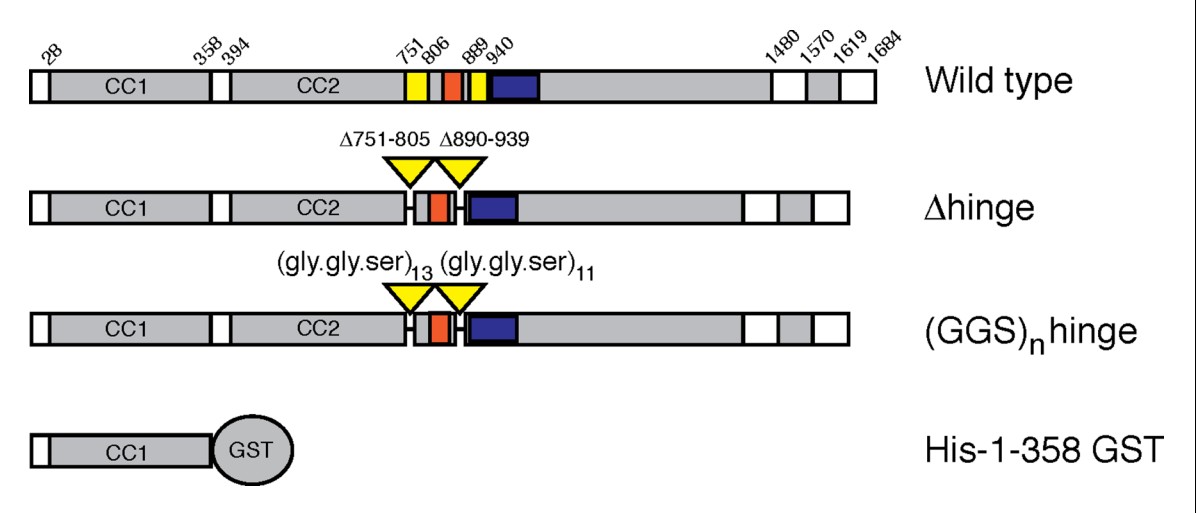

**Figure 1.** GCC185 rescue constructs tested. Top, Wild type; second row, △hinge mutant (residues 751– 805 and 890–939 deleted); third row, (GSS)n hinge mutant with residues 751–805 and 890–939 (yellow bars) replaced with 13 and 11 repeats of Gly-Gly-Ser, respectively; proteins had either N-terminal Myc tags or GFP followed by a FLAG tag. Fourth row, His-1–358-GST used for vesicle capture. Orange and blue bars reflect a Rab9 GTPase or AP-1 binding site, respectively, as reported previously (**Brown et al., 2011**).

purified in the absence of dithiothreitol (**Figure 3—figure supplement 3**). Without DTT, GCC185 appeared rod-like.

Remarkably, half of the GCC185 molecules displayed central unwinding of the coiled coil structure (**Figures 2,3**). To our knowledge, such unwinding in a coiled coil has not been detected previously. The unwound 'bubble' region was less frequently observed in GCC185 △hinge protein preparations (**Figure 2C,D**) and thus not a general artifact of protein deposition on mica; the mean size of the bubble was ~27 nm in wild type versus ~17 nm in △hinge molecules, with a much sharper size distribution in the latter case, and only 20% of the △hinge molecules showing any bubble at all (**Figure 2C,E**). The normalized splay-to-splay distance (**Figure 2F**) provides information about the molecule's tendency to bend in this region; rigid (or elongated) molecules have a longer splay-to-splay distance (approaching a value of 1) than bent molecules. △hinge proteins all showed the longest mean, normalized splay-to-splay distances; using this metric, wild type molecules with bubbles were more bent than those without bubbles (**Figure 2F**). These data show that GCC185 is a flexible molecule and its tendency to bend is contributed by the so-called 'hinge' residues 751-805 and 890-939.

That the molecules detected by AFM represent individual dimers rather than end-to-end tetramers was determined by introduction of an HA epitope tag into the central hinge region (**Figure 4A**); antibody labeling of such molecules was readily detected by AFM and showed antibodies at the position expected for the epitope tag location within a parallel dimer (**Figure 4B-D**). These data also confirm the location of the bubble in relation to the position of the epitope tag at residue 806.

AFM also revealed that purified GFP-FLAG-GCC185 has an overall mean length of ~145 nm (**Figure 3—figure supplement 1,2,4**). GCC185 fragments were also generated that span the length of the protein (residues 1-358, 394-751, 1-889 and 890-1684). Purified fragments spanning the first and second, predicted coiled-coils were similar in length; the N-terminal half of GCC185 (1-889) was longer than half the length of the complete protein (mean length = 91.2 nm; **Figure 3— figure supplement 2, 4**). The sum of the lengths of GCC185 fragments could account for the length of the full-length protein determined by AFM. However, full-length GCC185 and all fragments were ~30% shorter than the predicted lengths for coiled-coils with the same number of residues (the cartoons in **Figure 3—figure supplement 1,2** are drawn to scale for each portion of the molecules and summarized in **Figure 3—figure supplement 4**). These data indicate that GCC185's N-terminus contains

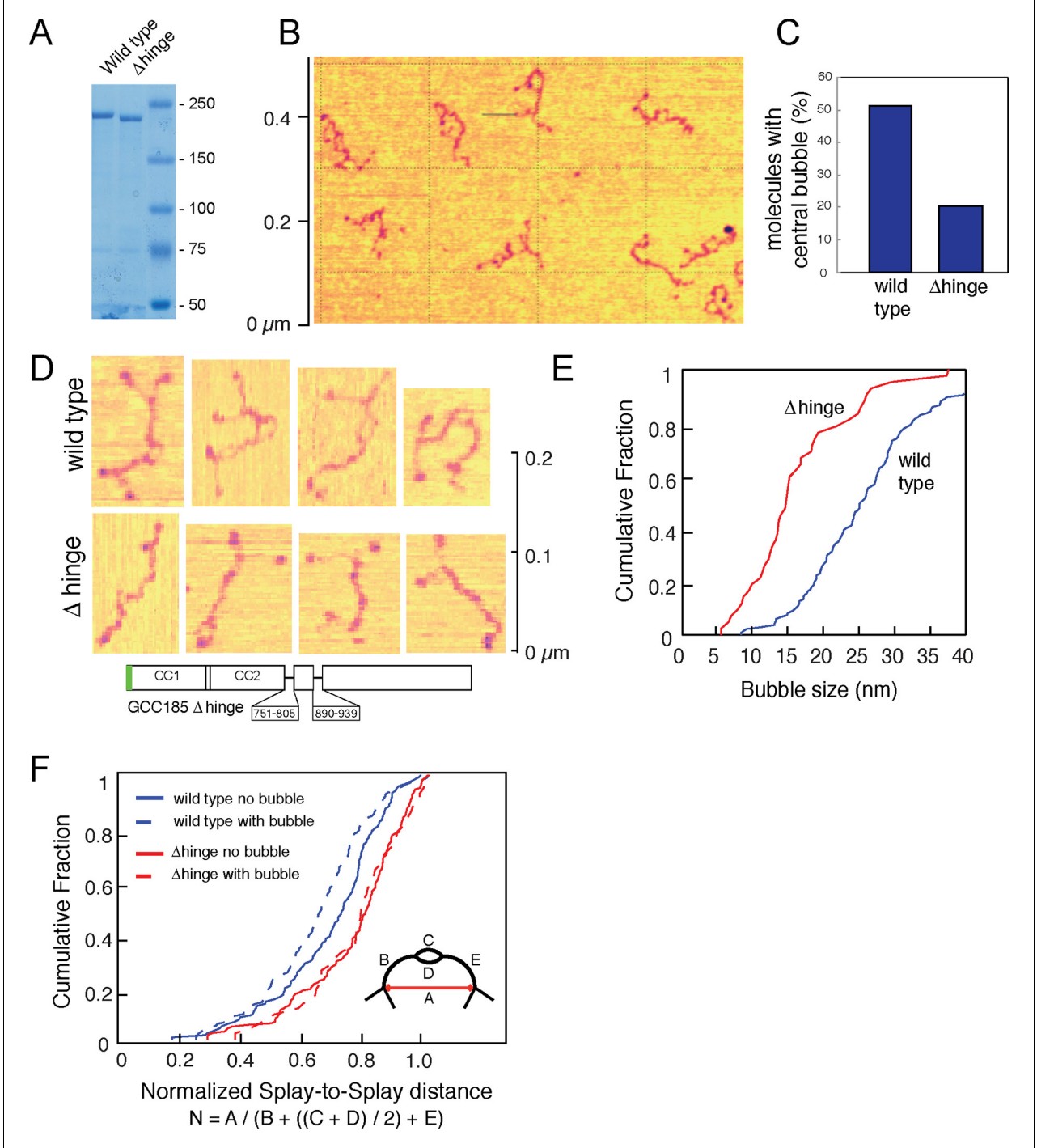

**Figure 2.** Atomic Force Microscopy reveals the splayed and flexible structure of GCC185. (**A**) Coomassie stained SDS-PAGE of indicated, purified proteins. (**B**) GFP-FLAG-GCC185 on mica and imaged in air from a 1.2 μm X 1.2 μm scan; Z range, 2.5 nm. (**C**) Frequency of bubbles in GCC185 and Δhinge molecules. (**D**) GFP-FLAG-GCC185 (top) and Δhinge mutant (bottom) molecules. E,F, Comparison of bubble size (**E**) and normalized splay-to-splay distance (**F**) of indicated molecules. Bubble size = (X4 + X5)/2 (see **Figure 3**); Normalized distance is indicated by the formula below and inset in f. Data in E,F were from 262 wild type and 204 Δhinge molecules (diagrammed below D). All the molecules measured were pooled from at least 3 independent experiments.

**Figure 3.** Conformations of GCC185's full length and N-terminal half. Column 1, Examples of different conformations seen for indicated GCC185 molecules visualized by AFM; Column 2, cartoon representations; Column 3, Relative abundance of each conformation. Total objects counted, full length, 262; residues 1–889, 350. Column 4, Segmentation of GCC185 for length and feature analyses. Data were pooled from at least 3 independent experiments.

The following figure supplements are available for figure 3:

**Figure supplement 1.** Map of the structural features of GCC185 wild type and △hinge mutant.

**Figure supplement 2.** Map of the structural features of GCC185 1-889.

**Figure supplement 3.** Initial comparison of the frequency of N-terminal splays and central bubbles in GCC185 or GCC185 1-889 purified in buffer A (50 mM Tris, 250 mM NaCl, 10% glycerol) ± 0.5mM DTT.

**Figure supplement 4.** Diagram showing actual fragment lengths compared with predicted lengths of the indicated constructs.

more coiled coil than the C-terminal half, and the entire protein contains less total coiled-coil structure than originally predicted.

To investigate the functional importance of the bubble-forming sequences (and their need to flex) for GCC185's tethering function, we tested the ability of the hinge deletion mutant (*Figure 1*) to rescue transport of MPRs from late endosomes to the Golgi in GCC185-depleted cells. In these experiments, immunoblots confirmed ~85% depletion of endogenous GCC185 protein (*Figure 5B* inset).

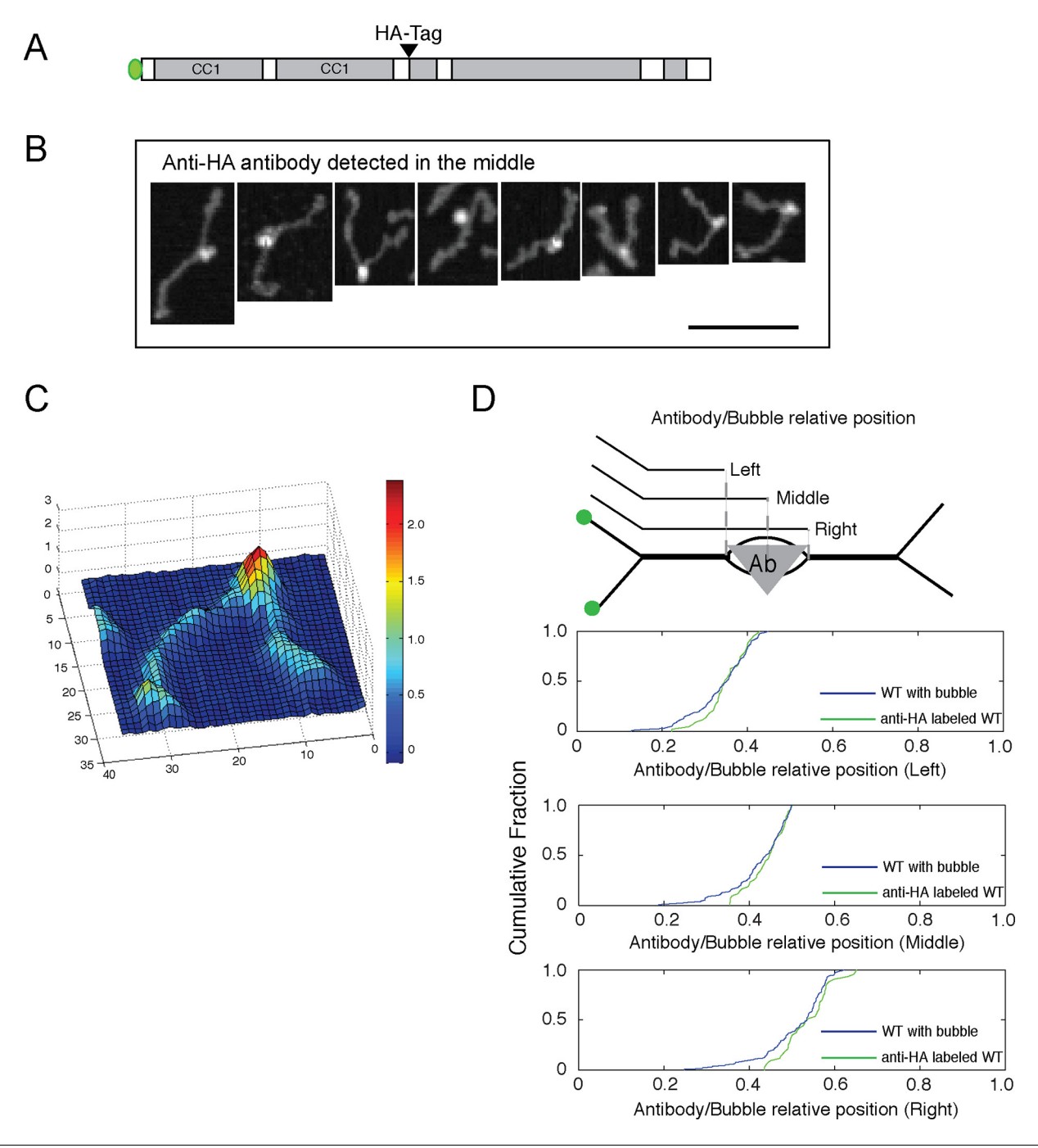

**Figure 4.** Antibody-labeling confirms the location of the central bubble. (**A**) Location of HA-tag after residue 805 in GFP-FLAG-GCC185. (**B**) Topographic AFM micrographs showing GFP-FLAG-GCC185 labeled with anti-HA antibody. Bar, 100 nm. (**C**) 3-D heat map of the height of a representative molecule with antibody bound. Rainbow scale, Z- range. (**D**) Relative bubble and antibody positions assessed by measuring the length from GFP to the beginning (left), center (middle) or the other end (right) of the bubble/antibody length as function of total length. Cumulative fraction plots of relative positions represent measurements of 43 antibody-labeled molecules (green) and 134 bubble-containing molecules (blue). Antibody-labeled molecules were imaged from 2 independent experiments.

As expected, in depleted cells, MPRs were detected in numerous, peripheral transport intermedi-ates; the Golgi ribbon was also dispersed (*Figure 5A,B*; *Hayes et al., 2009*; *Reddy, 2006*; *Brown et al., 2011*). This phenotype was readily reversed in cells co-expressing an siRNA-resistant

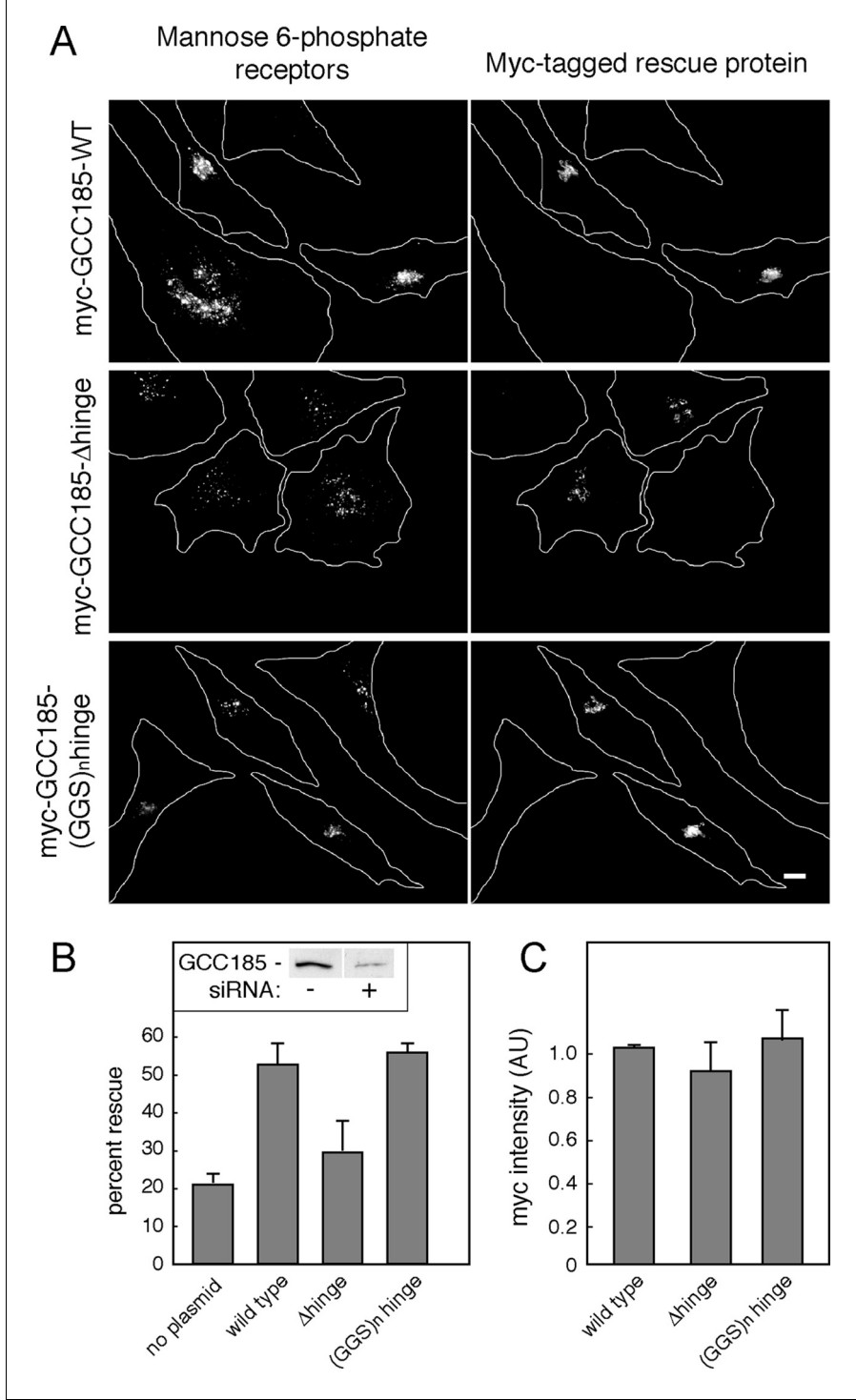

**Figure 5.** GCC185's flexible hinge region is required for receptor trafficking to the Golgi. (**A**) Left column, cation independent mannose 6-phosphate receptor localization detected using 2G11 mouse antibodies. Right column, expression of the indicated rescue constructs detected using chicken anti-Myc antibodies. Cell outlines are indicated. Scale bar, 10 μm. (**B**) inset, immunoblot of GCC185 ± siRNA treatment; Quantitation of rescue experiments ($p<0.05$ by one-way ANOVA). (**C**) Quantitation of Myc-tagged rescue protein levels using CellProfiler analysis of cells scored from light micrographs. Data in (**B**) and (**C**) represent the mean of three independent experiments; > 60 cells were counted for each condition in each experiment. Error bars represent standard deviation.

Myc-GCC185 protein (*Figure 5A,B*). In contrast, much less rescue was detected when the △hinge mutant protein, expressed at the same level (*Figure 5C*), was tested for rescue capacity (*Figure 5A, B*). Moreover, a protein in which putative hinge sequences were replaced with unstructured gly-gly-ser repeats of the same length (*Figure 1*) was fully capable of rescuing both transport vesicle receipt and normal Golgi morphology (*Figure 5A,B*). Quantification verified that all three myc-tagged, rescue constructs were expressed at similar levels in the individual, scored cells (*Figure 5C*; overall rescue construct transfection efficiency 60%), at no more than 3-5 fold the level of the endogenous protein for the total pool of cells. These experiments demonstrate the importance of hinge sequences for GCC185-mediated vesicle tethering, and show that these sequences provide flexibility rather than a binding site for another cellular component.

The above experiments show that GCC185 tethering requires central flexibility, implying that GCC185 may flex in the middle—to bring N-terminally bound cargo closer to the target membrane (and to the tether's C-terminus). This was tested using a 'proximity ligation assay' (PLA) to monitor the proximity of GCC185's N- and C-termini on the surface of the Golgi in cells. Briefly, cells are labeled with primary antibodies from different species that are specific for the two domains to be tested; secondary antibodies are added that have short oligonucleotides attached. A circularizing oligonucleotide is added, ligated, and amplified; the product is detected by hybridization to a fluorescent oligonucleotide. Epitopes that are within 40 nm of each other are detected as a fluorescent spot. The total intensity of PLA signal per cell can then be quantified (*Carpenter et al., 2006*).

Using mouse anti-GCC185 N-terminus and rabbit anti-GCC185 C-terminus antibodies, PLA signal was readily detected on Golgi complexes of individual cells (*Figure 6A*). In contrast, no signal was obtained using mouse anti-Golgin 245 C-terminus and rabbit anti-GCC185 C-terminus antibodies, even though both proteins are localized to the trans Golgi network. Similarly, mouse anti-cis-Golgi GM130 antibodies yielded only a very weak PLA signal when used in combination with rabbit anti-GCC185 antibodies. The specificity of the anti-GCC185 antibodies was confirmed using purified GCC185 N- and C-terminal domain fragments (residues 1–358 or 1342–1684; *Figure 6E*). Quantification of the PLA data confirmed specific PLA labeling for reactions with the GCC185 N- and C-terminal domain antibodies (*Figure 6B*), consistent with proximity of the N- and C-termini of this protein on the Golgi in cells.

We also tested the relative ability of △hinge GCC185 to yield a proximity signal for its N- and C-termini (*Figure 6C,D*). Cells expressing GFP-GCC185 wild type and GFP-GCC185 △hinge constructs showed similar levels of protein expression (*Figure 6D*, X-axis). However, GFP-GCC185-△hinge yielded a significantly weaker PLA signal when compared with cells expressing GFP-GCC185 (*Figure 6C,D*). This suggests strongly that the flexibility-conferring regions facilitate the ability of the N- and C-termini to achieve proximity on the Golgi. Quantitation of PLA signal intensity as a function of protein expression confirmed the importance of the hinge region for proximity ligation (*Figure 6D*). Only at higher expression levels was proximity detected for the GCC185-△hinge construct; at high expression levels, these are likely to represent inter- rather than intra-molecular processes. These experiments show that the presence of a flexible region is needed for GCC185 function and permits the N- and C-termini to become closer together, on the Golgi, in cells. It should be noted that proximity ligation could indicate bending/flexing of individual proteins or inter-digitation of two different proteins in anti-parallel orientation; we favor the former explanation, as it is hard to imagine that hinge deletion would influence the possibility of inter-digitation.

## Vesicle capture by GCC185 splayed N-termini

Current models posit that a tethering protein should capture vesicles at one end and be anchored to the target membrane at the other end. For GCC185, the N-terminus would therefore be predicted to bind vesicles. We used differential centrifugation to isolate a crude vesicle fraction carrying the cargoes, GFP-Rab9 (*Figure 7B*) and endogenous cation-independent MPR (CI-MPR; *Figure 7C*) and Rab9 (*Figure 7D*) proteins. We then compared the ability of GCC185 N-terminal residues 1-358 or (similarly sized) internal residues 1032–1331 (*Figure 7A*) to bind these vesicles after immobilization on glutathione-Sepharose. *Figures 7B-D* show that vesicles were indeed captured by the N-terminal 1–358 fragment but not by the internal fragment of similar length. Interestingly, the 1–358 sequences tethered most effectively when the obligate GST dimer was located at the fragment's C-terminus (1–358-GST) rather than the N-terminus (GST-1–358; *Figure 7C*). Specificity control experiments showed that neither early endosome (EEA1) nor endoplasmic reticulum (calnexin) markers

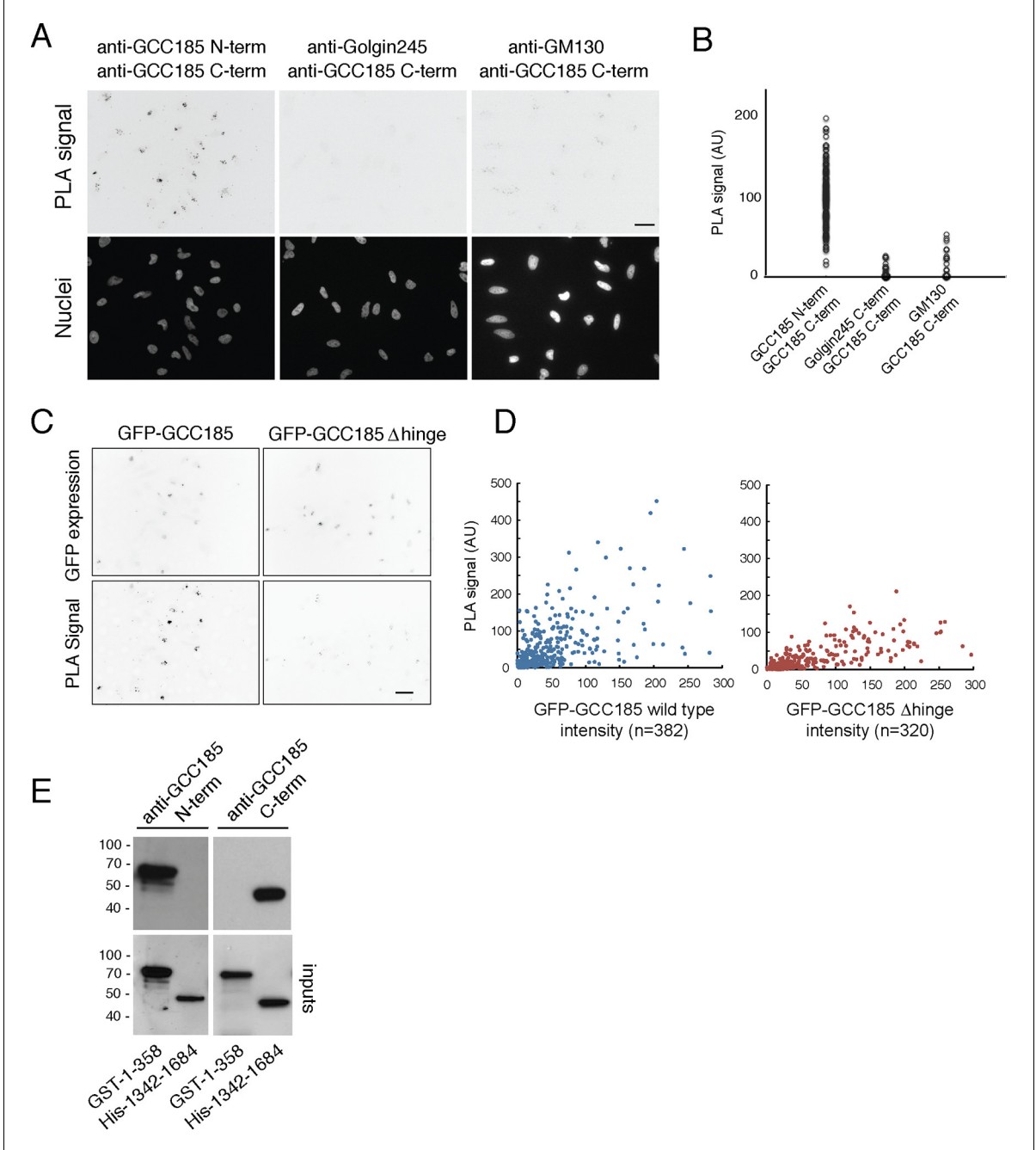

**Figure 6.** Proximity of GCC185 N- and C-termini on the Golgi requires putative hinge sequences. (**A**) Top: Proximity ligation in HeLa cells using indicated antibodies; nuclear staining is shown below. (**B**) Quantitation of proximity ligation using CellProfiler; N = 168, 94 or 46, left to right (*p*<0.0001 by two-tailed Student's t-Test). (**C**) Bottom: Proximity ligation using mouse-anti-GFP and rabbit-anti-GCC185 C-terminus antibodies in GFP-GCC185 or GFP-GCC185-Δhinge-transfected HeLa cells. Top, GFP-GCC185 expression in the same cells. Bar, 20 μm. (**D**) Quantitation of proximity ligation versus GFP-protein expression levels using CellProfiler. More than 300 objects were measured for each condition (*p*<0.001 by two-tailed Student's t-Test), pooled from two independent experiments. (**E**) Immunoblot test of antibody specificity using purified GCC185 domains. Upper panel: 10 ng GST-1–358 and His-1342–1684 were detected with (left) mouse anti-GCC185-N-term or (right) rabbit anti-GCC185-C-term antibodies. Lower panel: blots were re-probed using mouse anti-His (left) or rabbit anti-GST (right) antibodies to detect the corresponding antigens.

were captured by the constructs tested (*Figure 7C*). N-terminal sequences are essential for GCC185 function in cells (*Hayes et al., 2009*), consistent with their ability to bind transport vesicles in vitro.

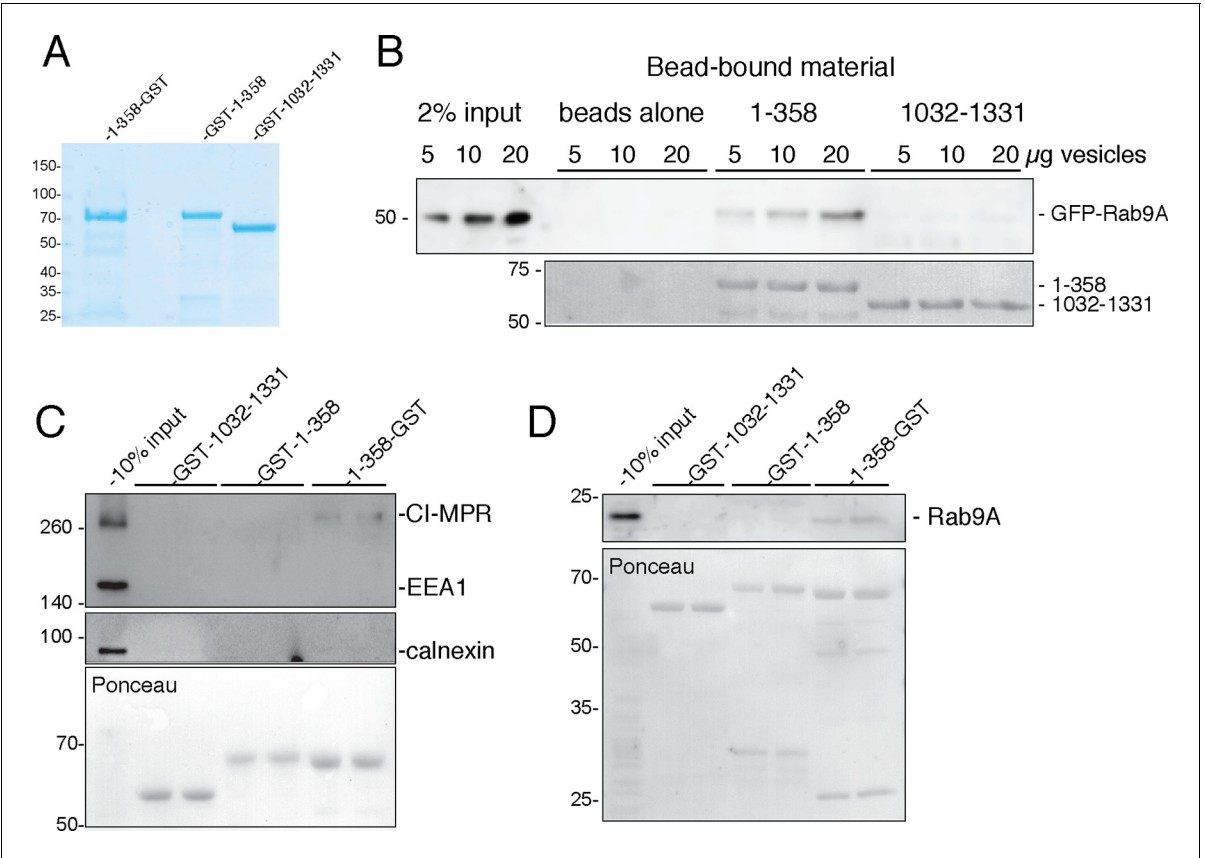

**Figure 7.** GCC185 N-terminal domain tethers cargo-containing vesicles. (**A**) Coomassie blue stained SDS-PAGE of purified His-1–358-GST, GST-1–358 and GST-1032–1331. (**B**) Binding of GFP-Rab9-vesicles to the indicated, immobilized GCC185 constructs. Top panel, anti-GFP immunoblot; bottom panel, Ponceau S stain. (**C,D**) Binding of endogenous CI-MPR or Rab9-vesicles, respectively, to the indicated constructs as in **B**. Top panels, **C**, anti-CI-MPR, anti-EEA1 and anti-calnexin blots; (**D**) anti-Rab9A blot; (**C,D**) bottom panels, Ponceau S staining. Mobility of marker proteins is indicated at left in Kd. A representative example from at least 3 independent experiments (in duplicate) is shown for GFP-Rab9- and endogenous CI-MPR-vesicle capture; 2 independent experiments (in duplicate) were performed for endogenous Rab9-vesicle capture.

The functional significance of N-terminal splaying detected in 80% of purified GCC185 molecules (*Figure 2,3*) is confirmed by the finding that the fragment capable of splaying at the N-terminus (1–358-GST) showed highest tethering activity (*Figure 7C,D*). Although we cannot rule out completely, the possibility of GST-steric hindrance for vesicle binding when present at the N-terminus, these data suggest that free N-termini represent both the active and most abundant form of this tethering protein. These experiments show that GCC185 N-terminal sequences can bind cargo—essential for a bona fide tethering protein. Moreover, this represents the first direct demonstration of in vitro vesicle tethering by a TGN localized, GRIP-domain containing tethering protein.

## Discussion

We have shown, for the first time, that flexibility in a tethering protein is required for its functionality in cells, both in supporting the receipt of transport vesicles at the Golgi and maintaining Golgi ribbon structure. We have shown that transport vesicles can bind to GCC185's N-terminus, and seem to prefer to bind to splayed ends of this dimeric, coiled coil tether. The presence of multiple Rab-binding sites along the length of individual Golgin proteins (*Sinka et al., 2008*; *Hayes et al., 2009*; *Short et al., 2001*) has led others to suggest that vesicles may hop along individual tethers (*Barr and Short, 2003*) or be captured by Golgin 'tentacles' within a potential Golgin meshwork towards the target membrane (*Munro, 2011*). Such a model considers tethers as rigid, extended rods (*Figure 8*, left). Our data favor a model whereby GCC185 is C-terminally anchored to the TGN

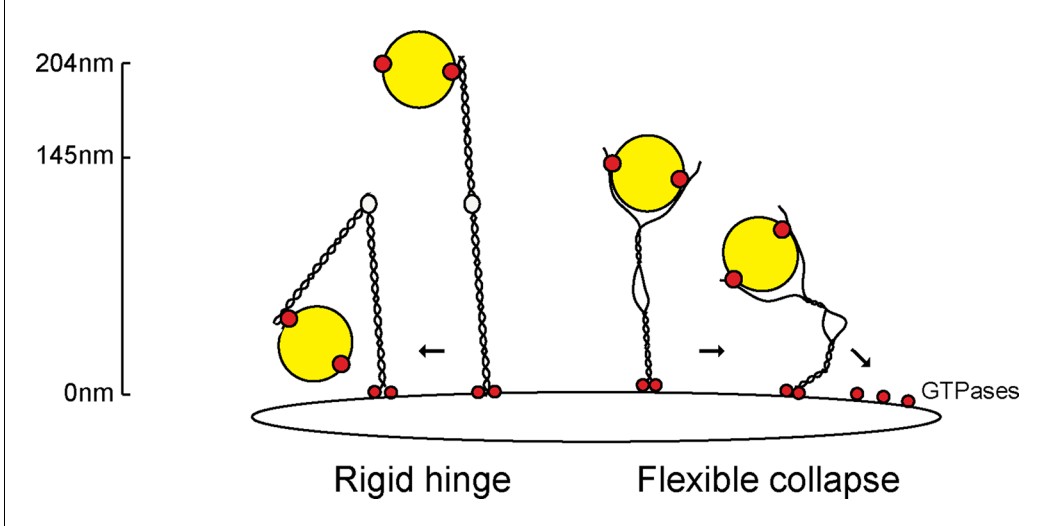

**Figure 8.** Collapse model for vesicle tethering at the TGN. Previous models based on GMAP-210 have led to a model that tethers extend from the Golgi as highly rigid rods that may bend in the middle ('Rigid hinge'). Data presented here show that GCC185 is shorter than originally predicted (145 nm vs. >204 nm), has splayed ends that can capture vesicles, and an unusual central bubble and floppiness as detected by atomic force microscopy. These features lead us to propose a flexible collapse model whereby the tether may collapse onto the Golgi surface, possibly facilitated by Rab GTPases and other proteins that are localized there. Note that vesicles may also (or alternately) bind near the bubble, and engage both the AP1 and Rab9 binding sites there, in addition to GCC185's N-terminal arms.

through Rab6 and Arl1 GTPases (*Burguete et al., 2008*) and captures Rab9-decorated transport vesicles (*Hayes et al., 2009*; *Reddy, 2006*) by binding them at the N-terminus and/or near the hinge region (*Brown et al., 2011*); importantly, the central bubble gives significant flexibility to the protein and would allow it to collapse onto the Golgi surface, bringing vesicles close to the trans Golgi target membrane (*Figure 8*). GCC185 can bind multiple Rab GTPases along its length (*Hayes et al., 2009*), as well as syntaxin16, Arl1 and Arl4 GTPases (*Burguete et al., 2008*; *Ganley et al., 2008*; *Lin et al., 2011*). GCC185 also binds to CLASP proteins that catalyze microtubule polymerization from the Golgi surface (*Efimov et al., 2007*). These additional interactions will all pull the tether towards the Golgi surface, bringing N-terminally bound vesicles closer to the Golgi surface for productive vesicle docking.

We have shown here that the functionally essential, GCC185 N-terminus has the capacity to bind to transport vesicles in vitro. N-terminal vesicle binding is shared with GMAP-210, a Golgin localized to the cis-Golgi (*Sato et al., 2015*); for other Golgins, it is presumed due to binding sites for various coat proteins. In previous work (*Brown et al., 2011*), we showed that GCC185 residues 938–1032 comprise an AP-1 binding site that is also needed for GCC185 vesicle tethering (blue bar in *Figure 1A*); the transport vesicles are AP-1 decorated, thus this was presumed to represent a possible vesicle binding site. This AP-1 binding site is located immediately downstream of the regions deleted in the △hinge protein studied here (751-805, 890939) (*Figure 1A*). Moreover, an apparently dispensable, Rab9 GTPase binding site is located directly between the two deletions made to generate the △hinge mutant (orange bar, *Figure 1A*). Binding of Rab9 to that site could influence the ability of that part of the protein to form a bubble, and could influence the efficiency of tethering on a time scale finer than that assayed here and previously. Both Rab9 and AP-1 are present on the transport vesicles captured, but we do not yet know the precise molecular interaction underlying the ability of the N-terminal domain to bind vesicles. That a splayed N-terminus is utilized suggests that the tether gains efficiency by avidity, and can reach out to 'hug' the vesicle. How the various binding sites along GCC185 coordinate interactions with the vesicle and the target membrane to accomplish vesicle tethering represents an important area for future work.

Given the presence of AP-1 and Rab9 binding sites adjacent to the central bubble (*Figure 1A*) and the additional capacity of the splayed N-terminus to capture transport vesicles, it is also possible that rather than being held near GCC185's N-terminus, the vesicle could bind primarily *near the*

*bubble* while still being enwrapped by the N-terminal arms. The model in *Figure 8* is drawn to scale and assumes that the transport vesicles are ~50 nm in diameter (*Barbero, 2002*); they need not be larger to be able to occupy both potential vesicle binding domains. The AP-1 binding site (939–1,031)is essential for GCC185's role in vesicle tethering but not Golgi ribbon maintenance (*Brown et al., 2011*), and must be considered in any model of GCC185 vesicle tethering.

Like GCC185, the other human GRIP domain Golgins are also predicted to contain a high proportion of coiled coil sequences. Golgin-245 shares with GCC185 two very obvious breaks in the coiled coil that may also form a central bubble; its absolute N-terminus is also likely splayed. The shorter Golgin-97 and GCC88 proteins are likely about half as long as GCC185 by sequence alone; Golgin-97's N-terminus is likely splayed, and GCC88 may very well also contain a central bubble.

Finally, a domain between GCC185 residues 1332 and 1438 is required for maintenance of Golgi ribbon structure but not transport vesicle tethering (*Brown et al., 2011*). This observation highlights the fact that workers in this field have oversimplified thinking about how tethers function—simple engagement of N- and C-termini is not sufficient for both vesicle capture and Golgi ribbon stabilization. A recent paper studying Golgi tethers (*Wong and Munro, 2014*) concluded that unlike other TGN tethers, GCC185 could not tether vesicles upon re-localization to mitochondria. The extent of Golgi membrane mobilization seen in that study was far greater than the small pool size of the much rarer, Rab9-containing, AP-1 positive, mannose phosphate receptor transport vesicle intermediates—that are only visualized in cells upon depletion of tethering factors (*Reddy, 2006*; *Brown et al., 2011*). It is very possible that that study assayed the ability of TGN tethers to recruit/assemble TGN ribbons, rather than capture rare transport vesicle intermediates. Since ribbon assembly and transport vesicle capture are driven by separate domains of GCC185, perhaps GCC185 is more potent in vesicle tethering rather than Golgi ribbon assembly.

A remaining mystery is why depletion of any one of the GRIP domain-containing Golgins leads to Golgi fragmentation. Whatever the explanation, the floppy appearance of purified GCC185, when deposited on mica in the absence of fixative, and the proximity of GCC185's N- and C-termini on the Golgi in cells, argue strongly against this protein behaving like a pipe cleaner, protruding on the surface of the trans Golgi network in cells. Instead, tether collapse represents an appealing, alternative model.

## Materials and methods

### Expression plasmids

cDNAs encoding GCC185 full-length and fragments were prepared by ligating the full-length sequence and sequence encoding residues 1–358, 394–751, 1–889, 890–1684 into EGFP-C1 with a FLAG tag (DYKDDDDK) inserted downstream of GFP flanked by the GGATCC linker (AS) for purification purposes. The hinge mutant constructs were prepared by introducing GAGCTC (EL) at residue 751 and 805, and CTCGAG (LE) at 890 and 939 to remove the original sequence. Sequential digestion with SacI and XhoI restriction enzymes (New England Biolabs, Ipswich, MA) followed by re-ligation of the plasmid or ligating GGS repeats to the cleaved site created the △hinge (1–750-EL-806–889-LE-940-1684) and $(GSS)_n$ hinge constructs (1–750-EL-$(GSS)_{13}$-EL-806–889-LE-$(GSS)_{11}$-LE-940–1684), respectively. The HA-fusion construct was prepared by inserting the HA tag sequence (Y-PYDVPDYA) at residue 805 flanked by linker nucleotide sequence GAGCTC (EL) using the same strategy (1–804-EL- YPYDVPDYA-EL-807–1684). cDNAs encoding rescue plasmids were prepared with 8 silent mutations in the siRNA-targeted region and ligated into pcDNA3.1(+) (Invitrogen, Grand Island, NY) modified with an N-terminal Myc tag as described (*Hayes et al., 2009*). Bacterial expression plasmids were prepared by ligating PCR products encoding residues 1–358 or 1032–1331 into pGEX4T-1 (GE Healthcare, Pittsburgh, PA) (GST-1–358, GST-1032-1331) or using the pQE60 vector (Qiagen, Hilden, Germany) with a 6X His tag upstream and a GST tag downstream of residues 1–358 (His-1–358-GST). Residues 1342–1684 were inserted into the 6X-His fusion vector pET28a (Stratagene, La Jolla, CA).

### Cell culture and transfections

HeLa cells and HEK293T cells were cultured at 37°C and 5% $CO_2$ in Dulbecco's modified Eagle's medium (GIBCO, Life Technologies, Grand Island, NY) supplemented with 7.5% fetal bovine serum,

100 U/ml penicillin and 100 µg/ml streptomycin. Transfection with siRNA and rescue plasmids was performed as described (*Brown et al., 2011*). In brief, cells were transfected with GCC185 siRNA targeting the sequence 5′-GGAGTTGGAA CAATCACAT-3′ using Oligofectamine (Invitrogen) according to the manufacturer. For rescue experiments, depletion was for 72 hr; Myc -tagged rescue constructs resistant to the siRNA were transfected 24 hr after initial siRNA treatment using FuGENE6 (Roche, Indianapolis, IN). For GFP-Rab9 membrane tethering assay, GCC185-depleted 293T cells grown to 50% confluency were transfected with GFP-Rab9 construct construct using polyethyleni-mine (Polysciences Inc., Warrington, PA) for 24 hr before the extraction of membranes. Endogenous vesicle tethering assays were preformed with non-depleted 293T cells grown to 80–90% confluency. For proximity ligation assays, HeLa cells grown to 50% confluency were transfected with GFP-fusion constructs for 24 hr before fixation.

## Immunofluorescence microscopy

Cell fixation, staining and mounting in Mowiol were performed as described (*Hayes et al., 2009*). In brief, cells were split onto 22 x 22-mm coverslips in a 6-well plate 48 hr before fixation. Cells were washed twice in PBS and fixed for 20 min in 3.7% formaldehyde in 200 mM Hepes, pH 7.4. Fixed cells were then washed twice in PBS and permeabilized for 5 min with 0.2% Triton X-100 in PBS followed by two washes with PBS and a 15-min blocking with 1% BSA in PBS. Monoclonal mouse anti-cation-independent MPR antibody (2G11; *Lombardi et al., 1993*) and chicken anti-Myc (1:1000, Bethyl Laboratories, Inc., Montgomery, TX) primary antibodies, and Alexa Fluor 488 goat anti-mouse (1:2000) and Alexa Fluor 555 goat anti-chicken (1:2000) secondary antibodies (Invitrogen) were used. Micrographs were acquired as previously described (*Brown et al., 2011*). In brief, images were acquired using a microscope (Eclipse 80i; Nikon, Tokyo, Japan) fitted with a 60x/NA 1.4 plan apochromat objective lens, a Sedat Quad filter set (Chroma Technology Corp., Bellows Falls, VT) and a charge-coupled device camera (CoolSnapHQ; Photometrics, Tucson, AZ) at room temperature. Wavelength selection was performed using a controller (Lambda 10–3; Sutter Instrument, Novato, CA). Z sections were acquired with a Z axis drive (MFC-2000; Applied Scientific Instrumentation, Eugene, OR) at 0.2-mm steps. All instrumentation was controlled by MetaMorph imaging software (Molecular Devices, Sunnyvale, CA). Image deconvolution was performed using a theoretical point spread function with softWoRx (v.4.1.0; Applied Precision Inc., Issaquah, WA). The deconvolution micrographs displayed in *Figure 5* represent projections of three to five central slices of the z stack acquired. Two individuals used blinded datasets to score rescue experiment images.

## Proximity Ligation Assay and signal analysis

The Duolink in situ proximity ligation assay (PLA, Olink Bioscience, Uppsala, Sweden) was performed according to the manufacturer. Briefly, HeLa cells transfected with GFP-GCC185 or GFP-GCC185-△ hinge were washed, fixed and blocked. Rabbit anti-GCC185 C-terminus (1:500, *Reddy, 2006*), mouse anti-GCC185-1-358 N-terminus (1:500), mouse anti-GM130 (1:500; BD Biosciences, San Jose, CA), mouse anti-Golgin 245 (1:500; BD Bioscience) and mouse anti-GFP (clones 7.1 and 13.1; 1:100; Roche) primary antibodies were used. After 1 hr primary antibody incubation at room temperature, the PLA probe anti-mouse Plus and anti-rabbit Minus were diluted in PBS containing 1% BSA and added to cells for 1 hr at 37°C in a humidified chamber. All subsequent steps were performed at 37°C in a humidified chamber. Cells were then incubated with oligonucleotides complementary to the proximity probe and T4 DNA ligase for 30 min. Rolling-circle amplification was initiated by addition of Phi29 DNA polymerase and Texas red-labeled nucleotide probe. After 100 min, cells were washed and mounted and imaged using an Axiophot2 epifluorescence microscope fitted with a Plan-Neofluar 40x/0.75 objective (Carl Zeiss MicroImaging, Thornwood, NY), and a digital CCD camera ORCA-R[2] C10600 (Hamamatsu, Hamamatsu City, Japan). All microscope instrumentation was controlled by AxioVision4.7.2 software (Carl Zeiss MicroImaging). Individual cells (objects) were identified, and proximity ligation and GFP signals were quantified using CellProfiler (Broad Institute, Cambridge, MA; www.cellprofiler.org).

## Protein expression and purification

Expression and purification of GST-tagged constructs were as described (*Brown et al., 2011*). Constructs were transformed into BL21 (DE3) RIPL competent cells and bacterial cultures were induced

at $OD_{600}$ = 0.5-0.7 with 0.25 mM IPTG for 3 hr at 30°C. Cells were resuspended in 50 mM Hepes, pH7.4, 250 mM NaCl, 10% (v/v) glycerol and protease inhibitors and lysed by two passes through an EmulsiFlex-C5 apparatus (Avestin, Ottawa, ON) at >10,000 lb/in² followed by centrifugation at 13,000 rpm for 30 min in a FiberLite F15 (8x50c) rotor (Thermo Scientific). Clarified lysates containing GST-tagged proteins (His-1-358-GST, GST-1-358 and GST-1032-1331) were incubated with glutathione Sepharose resin (GE Healthcare) for 1 hr at 4°C. Bound GST fusion proteins were eluted in buffer containing 20 mM glutathione. GST-1-358 and GST-1032-1331 were loaded onto PD MiniTrap G-25 (GE Healthcare) to remove glutathione. His-1-358-GST protein was further incubated with Ni-NTA agarose (Qiagen) for 2 hr at 4°C in buffer containing 20 mM imidazole and eluted with 300 mM imidazole after 3 washes. Same amount of purified His-1–358-GST, GST-1–358 and GST-1032–1331 were immobilized on glutathione-Sepharose resin (3–5 mg protein/ml resin) in 50 mM Hepes pH7.4, 200 mM NaCl, 0.5 mM DTT and 10% glycerol. Resins were washed and equilibrated in 50 mM Hepes pH7.4, 150 mM NaCl, 5mM $MgCl_2$, 0.5 mM DTT and 50 µM GTP for subsequent membrane capturing assays. His-tagged 1342–1684 was incubated with His-Pur Ni-NTA resin (Thermo Scientific) for 2 hr at 4°C and eluted with buffer containing 200 mM imidazole after 3 washes. Eluted protein was further purified by gel filtration on Superdex 75 (GE Healthcare) in 50 mM Tris, pH7.4 250 mM NaCl and 10% (v/v) glycerol.

GFP-FLAG-GCC185 constructs for AFM were produced by transfection of HEK293F mammalian suspension cells grown in FreeStyle 293 expression medium (GIBCO, Life Technologies) with the indicated constructs for 22 hr. Cells were lysed in 50 mM Tris, pH 7.4, 250 mM NaCl, 0.5% Triton X-100, 10% glycerol, 1 mM EDTA, 5 mM sodium orthovanadate and protease inhibitors. Lysates were clarified by centrifugation at 13,000 rpm for 30 min in a FiberLite F15 (8x50c) rotor (Thermo Scientific, Waltham, MA). Supernatants were incubated with anti-FLAG M2 affinity gel (Sigma Aldrich, St. Louis, MO) for 4 hr at 4°C, followed by washes with buffer containing 1 M NaCl and two final washes in 50 mM Tris, pH7.4, 250 mM NaCl and 10% glycerol. Bound proteins were eluted with 0.1 mg/ml 3X FLAG peptide (Sigma-Aldrich). Eluted proteins were further purified on a Sepharose 4B (Sigma-Aldrich) column. Anti-HA antibody labeling of the 805-HA construct was performed after FLAG peptide elution (Sigma-Aldrich). Excess anti-HA antibody (Abcam, Cambridge, MA) was added at 4°C for >2 hr. Excess antibodies were removed using Sepharose 4B.

## Membrane vesicle preparation and assay

At least three transfected 10-cm dishes (80–90% confluency) were harvested for vesicle isolation. Cells were washed with PBS and incubated in hypotonic buffer (10 mM Hepes pH7.4) for 10 min, scraped immediately into resuspension buffer (10 mM Hepes, pH7.4, 150 mM NaCl, 5 mM $MgCl_2$ + protease inhibitors) and further lysed by passage through a 25-G needle. Lysed cells were spun at 800 $g$ for 5 min at 4°C to remove nuclei and cell debris. The supernatant was spun at 135,000 $g$ for 15 min at 4°C in TLA100.2 rotor. The membrane pellet was resuspended in 50 mM Hepes pH7.4, 150 mM NaCl, 5 mM $MgCl_2$, 0.5 mM DTT and 50 µM GTP. The crude suspension was further spun at 8,000 $g$ for 5 min at 4°C to pellet large membranes; the supernatant was used as a crude vesicle fraction. Fifteen microliters GST-tagged protein-bound glutathione Sepharose resin was incubated with 100 µl of crude vesicle membranes (5–20 µg of GFP-Rab9-transfected cell or 50 µg non-transfected cell membrane) for 45 min at room temperature with rotation. Samples were washed with 100 column volumes of buffer and eluted with buffer containing 20 mM glutathione or sample buffer without reducing agent for CI-MPR immunoblot detection. Captured components were detected by immunoblotting with chicken anti-GFP (1:1000, Life Technologies), rabbit anti-CI-MPR (1:1000) mouse anti-EEA1 (1:1000, BD Biosciences), goat anti-Calnexin (C-20, 1:1000, Santa Cruz Biotechnology, Dallas, TX) and the corresponding horseradish peroxidase (HRP)-conjugated secondary antibodies: rabbit anti-chicken HRP (1:1500, Promega, Madison, WI), goat anti-rabbit HRP (1:3000, BioRad, Hercules, CA), goat anti-mouse HRP (1:2000, BioRad) and fluorescently-labeled bovine anti-goat DyLight 649 (1:2000, Jackson ImmunoResearch Laboratories, West Grove, PA). HRP antibodies were detected using ECL plus (ThermoFisher) in conjunction with film and Versadoc.

## Imaging GCC185 by atomic force microscopy

For immobilization, 25 µl purified sample (~1 nM) in 50 mM Tris, pH 7.4, 200 mM NaCl, 5 mM $MgCl_2$, 0.5 mM DTT and 10% glycerol was directly placed onto freshly cleaved mica. After 8

minutes, the mica surface was gently rinsed with 2 ml buffer followed by 2 ml of $ddH_2O$ to remove unbound proteins and dried by soft argon flow. Imaging was done in air with a Park100 AFM (Park Systems, Santa Clara, CA) operated in non-contact mode. We used ultra-sharp diamond-like carbon probes (NSG10 DLC, ~2–5 nm radius of curvature at the tip (K-Tek Nanotechnology, Wilsonville, OR) driven near resonance at ~200 kHz. To minimize both sample and tip damage during scanning, the set point was set to the largest possible value (largest average tip-sample distance) that allowed good surface tracking. Raster scans (1.2 μm X 1.2 μm) were performed at a resolution of 512x512 pixels with a scan rate of 1Hz. For presentation and analysis, images were first processed using a custom Matlab script (*Source code 1* provided) to remove sample slant (flattening) by row and column-wise median subtraction.

## Analysis of AFM images

Individual molecules of each type were selected manually based on the following criteria: the whole object should be within the captured field without any part(s) being cropped by the borders, easily identified as isolated, individual, non-aggregated object with no overlap(s) with any other objects, and traceable/measurable. The number of selected molecules usually covered 40–70% of all the objects in the field, depending on the distribution of molecules and local quality of an image. To quantify the conformation of each molecule, a selected molecule was segmented into maximally 8 sections based on apparent structural features. The length of each segment (X1 to X8) was measured with ImageJ by setting the measuring scale with the known image spatial size and outlining the molecule contour with the segmented line tool.

Statistics- Minimum sample sizes were determined assuming 5% standard error and >95% confidence level; P values and their methods of determination are indicated in the legends.

# Acknowledgements

This research was funded by a grant to SRP from the US NIH (DK37332). PC was the recipient of a fellowship from the American Heart Association Western States Affiliate; CL was supported by a Technology Development Grant from the Beckman Center at Stanford University.

# Additional information

### Competing interests

SRP: Reviewing editor, *eLife.* The other authors declare that no competing interests exist.

### Funding

| Funder | Grant reference number | Author |
|---|---|---|
| National Institute of Diabetes and Digestive and Kidney Diseases | DK37332 | Suzanne R Pfeffer |
| American Heart Association | | Pak-yan Patricia Cheung |
| Stanford University | | Charles Limouse |

The funders had no role in study design, data collection and interpretation, or the decision to submit the work for publication.

### Author contributions

PyanPC, Carried out the biochemical and cell biology experiments, discussed the results and wrote and/or commented on the final manuscript; CL, Carried out the AFM and was key to the analysis of AFM data, discussed the results and wrote and/or commented on the final manuscript; HM, Discussed the results and wrote and/or commented on the final manuscript; SRP, Contributed to the study design, discussed the results and wrote and/or commented on the final manuscript

### Author ORCIDs

Suzanne R Pfeffer, http://orcid.org/0000-0002-6462-984X

## Additional files

**Supplementary files**

• Source code 1. Matlab script for afm image analysis.

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
