## [Decision Letter]

Thank you for submitting your work entitled "Protein Flexibility is required for vesicle tethering at the Golgi" for consideration by *eLife*. Your article has been favorably evaluated by Randy Schekman (Senior Editor) and three reviewers, one of whom, Fred Hughson, is a member of our Board of Reviewing Editors.

The reviewers have discussed the reviews with one another and the Reviewing Editor has drafted this decision to help you prepare a revised submission.

Summary:

The authors investigate the Golgin protein GCC185, which has functions in Golgi morphology and endosome-derived vesicle tethering. They demonstrate that two central hinge regions of GCC185 are required for GCC185 function in cells, and that these hinges can be replaced by synthetic flexible linkers (Gly-Ser repeats) to restore function. Proximity ligation is used to demonstrate that the N- and C-termini of GCC185 can be located within 40 nm of each other (despite a predicted length of ~200 nm for a rigid GCC185 molecule). Importantly, this proximity required the hinge region.

Purified full-length GCC185 was also analyzed by AFM. The authors observe a central bubble region that corresponds to the region containing the hinges. They find that the hinge region is required for enabling the molecule to form more compact structures. The authors also find that the N-terminus in particular exhibits splaying in which the two monomers are not coiled together. This is interesting because the authors also show that that the N-terminus of GCC185 can capture Rab9/CI-MPR vesicles from a crude vesicle fraction. Interestingly, the orientation of this N-terminus fused to GST is important – the more physiological chimera performs best.

This manuscript addresses a number of important and longstanding questions in the field of membrane traffic. We enjoyed reading the manuscript and felt that it is the type of exciting and pioneering study that *eLife* should publish. It certainly leaves many questions unanswered, but it also provides a real advance for an important area in cell and membrane biology.

Essential revisions:

1) The data and the model indicate that the GCC185 N-terminus is most important for capturing incoming Rab9 vesicles. However, the established Rab9 and AP-1 binding sites are both near the two hinges. This central region seems like the least likely part of the protein to be able to contact the Golgi. How do the authors reconcile these disparate facts? Is it possible that the size of the vesicle comes into play (how large are Rab9/CI-MPR vesicles)? In Figure 8, is the size of the vesicle shown to scale relative to the GCC185 molecule (which the authors have now measured precisely)? Some discussion of these issues is merited and should be included in a revised manuscript.

2) The authors should seriously consider re-ordering the manuscript, with the AFM data first. One point of view is that the story is essentially backwards: the AFM data in Figure 5–Figure 7 allows one to make sense of the cell function data in Figure 1–Figure 4. Combining Figure 1 and Figure 5 into a new Figure 1, then moving the data in Figure 2–Figure 4 to the end of the manuscript would focus attention on the most important part of the work.

3) Can the authors exclude that flexibility is not required for initial vesicle tethering but is required for the tethering cycle to complete and for the vesicle to fuse? In other words, might they be stalling the tethering cycle with the hinge mutation rather than blocking tethering? The functional data doesn't have the resolution to pick that up. One option might be to change the abstract to say "that flexibility in a specific, central region is required for GCC185's ability to function in a vesicle tethering cycle."

4) Figure 2 needs some western blot controls for depletion of endogenous GCC185 and to estimate the level of replacement using various GCC185 transgenes once transfection efficiency is accounted for.

---

## [Author Response]

*Essential revisions:*

*1) The data and the model indicate that the GCC185 N-terminus is most important for capturing incoming Rab9 vesicles. However, the established Rab9 and AP-1 binding sites are both near the two hinges. This central region seems like the least likely part of the protein to be able to contact the Golgi. How do the authors reconcile these disparate facts? Is it possible that the size of the vesicle comes into play (how large are Rab9/CI-MPR vesicles)? In Figure 8, is the size of the vesicle shown to scale relative to the GCC185 molecule (which the authors have now measured precisely)? Some discussion of these issues is merited and should be included in a revised manuscript.*

The actual size of the transport vesicle intermediates is not known and we have assumed it is ~50 nm. We thank the referees for asking us to make sure the model is drawn to scale (it needed slight revision). We have added new text to the Discussion to consider these points and redrawn Figure 8 to scale and added a Figure 8 text legend. We think it is likely that the tether can in fact collapse down onto the Golgi surface.

“Given the presence of AP-1 and Rab9 binding sites adjacent to the central bubble (Figure 1) and the additional capacity of the splayed N-terminus to capture transport vesicles, it is also possible that rather than being held near GCC185’s N-terminus, the vesicle could bind primarily near the bubble while still being enwrapped by the N-terminal arms. […] The AP-1 binding site (939–1,031)is essential for GCC185’s role in vesicle tethering but not Golgi ribbon maintenance (Brown, Schindelhaim, and Pfeffer, 2011), and it must be included in any model of vesicle tethering.”

*2) The authors should seriously consider re-ordering the manuscript, with the AFM data first. One point of view is that the story is essentially backwards: the AFM data in Figure 5–Figure 7 allows one to make sense of the cell function data in Figure 1-4. Combining Figure 1 and Figure 5 into a new Figure 1, then moving the data in Figure 2–Figure 4 to the end of the manuscript would focus attention on the most important part of the work.*

We have re-ordered the paper exactly as requested. We kept Figure 1 as separate as not to minimize the size of the AFM images in Figure 2. In an electronic journal, this should not be a problem.

*3) Can the authors exclude that flexibility is not required for initial vesicle tethering but is required for the tethering cycle to complete and for the vesicle to fuse? In other words, might they be stalling the tethering cycle with the hinge mutation rather than blocking tethering? The functional data doesn't have the resolution to pick that up. One option might be to change the abstract to say "that flexibility in a specific, central region is required for GCC185's ability to function in a vesicle tethering cycle."*

We agree completely and have changed the Abstract for better clarity.

*4) Figure 2 needs some western blot controls for depletion of endogenous GCC185 and to estimate the level of replacement using various GCC185 transgenes once transfection efficiency is accounted for.*

We have added a western blot inset to the new Figure 5; the data came from light micrographs where we are certain that the construct is actually expressed, as determined by Myc-antibody staining, all quantified by CellProfiler to be sure that we are looking at cells with comparable levels of myc-construct expression (Figure 5). We estimate that for the total population of cells, the exogenous rescue proteins are expressed at no more 3-5 times the level of the endogenous protein (Results, seventh paragraph) – but the direct, single cell evaluation is most accurate.